# CodeScore: Evaluating Code Generation by Learning Code Execution

## Abstract

A proper code evaluation metric (CEM) profoundly impacts the evolution of code generation, which is an important research field in NLP and software engineering. Prevailing match-based CEMs (e.g., BLEU, Accuracy, and CodeBLEU) suffer from two significant drawbacks. 1. They primarily measure the surface differences between codes without considering their functional equivalence. However, functional equivalence is pivotal in evaluating the effectiveness of code generation, as different codes can perform identical operations. 2. They are predominantly designed for the Ref-only input format. However, code evaluation necessitates versatility in input formats. Aside from Ref-only, there are NL-only and Ref&NL formats, which existing match-based CEMs cannot effectively accommodate. In this paper, we propose CodeScore, a large language model (LLM)-based CEM, which estimates the functional correctness of generated code on three input types. To acquire CodeScore, we present UniCE, a unified code generation learning framework, for LLMs to learn code execution (i.e., learning PassRatio and Executability of generated code) with unified input. Extensive experimental results on multiple code evaluation datasets demonstrate that CodeScore absolutely improves up to 58.87% correlation with functional correctness compared to other CEMs, achieves state-of-the-art performance, and effectively handles three input formats.

## 1 Introduction

Automatic evaluation of code generation is significant and promising in the fields of natural language processing (NLP) and software engineering. Due to the great potential of code generation in reducing development costs and revolutionizing programming modes, both industry and academia have devoted substantial attention to it Li et al. (2022); Mukherjee et al. (2021); Yin and Neubig (2018); Chen et al. (2021); Shen et al. (2022); Dong et al. (2022). Code generation has achieved remarkable developments in the past few years Fried et al. (2022); Nijkamp et al. (2022); Dong et al. (2023a); Jiang et al. (2023), but CEMs still need to catch up. It is challenging to evaluate the competitiveness of various approaches without proper CEM, which hampers the development of advanced techniques for code generation. A range of code generation subtasks would benefit from valid code evaluation, including code completion Guo et al. (2022a); Lu et al. (2022), code translation Rozière et al. (2020); Zhu et al. (2022), code search Sun et al. (2022); Arakelyan et al. (2022), etc. Therefore, research on code evaluation is necessary and should be put on the agenda.

Some commonly used match-based CEMs treat code as text, such as BLEU Papineni et al. (2002) and Accuracy, which focus on basic and lexical-level features. They compute scores mainly based on n-gram co-occurrence statistics. CodeBLEU Ren et al. (2020) additionally takes into account the structure of code, i.e., abstract syntax tree and data flow. However, the preceding CEMs have deficiencies in identifying code relationships, because code is mainly evaluated based on functional correctness rather than exact/fuzzy match to reference code, and match-based CEMs cannot account for the large and complex space of code functionally equivalent to reference code Inala et al. (2022). For example, in Fig. 1, code (a) and code (b) have a much higher similarity of tokens or structures than code (c). But through execution, we realize that code (a) and code (c) are different renderings of the same function. By contrast, the execution result of code (b) differs dramatically from both other codes, and code (b) even fails to compile. As a result, merely measuring the similarity of token/structure is insufficient for code evaluation.

Figure 1: Results of evaluating the generated code implementing bubble sort using different CEMs. BLEU and CodeBLEU score the truly functional correct code (c) lower than the incorrect code (b).

LLMs pre-trained on code have demonstrated outstanding results in code generation tasks Chen et al. (2021); Fried et al. (2022); Li et al. (2022); Dong et al. (2023b), which are fundamentally dependent on exceptional code comprehension. Excellent code comprehension is a crucial element for facilitating code evaluation. We hypothesize that LLMs pre-trained on code possess the ability to evaluate code. However, due to the training strategy of predicting the next token according to context, they lack awareness of evaluating code for functional correctness. Our objective is to instruct LLMs to evaluate code effectively in terms of functional correctness.

Another issue that requires resolution is that the existing match-based CEMs are exclusively confined to the Ref-only input format. This restriction presents three inherent disadvantages. First, for any code generation task, the correct solutions are not finite, but rather, they are inexhaustible. In this context, the provided reference code merely represents one correct solution among a vast multitude. Therefore, it is overly narrow to compare the generated code solely with one correct solution. Second, they neglect the natural language (NL) description, which is a rich repository of information and a real requirement source. Third, these metrics are unusable in the absence of a reference code. This situation is quite commonplace in real-world evaluations where a correct solution is not always readily available. As a result, expanding the input format of CEM is necessary.

In this paper, we propose an effective LLM-based CEM, called CodeScore, which measures the functional correctness of generated codes on three input formats (Ref-only, NL-only, and Ref&NL). To obtain CodeScore, we present a code evaluation learning framework, UniCE, for tuning LLMs to estimate execution similarities with unified input. Specifically, we finetune LLMs to learn PassRatio and Executability of generated code, where Executability is devised to distinguish between compilation errors and output errors for code with PassRatio equal to 0. Generally, codes exhibiting higher functional correctness will pass more test cases, thereby achieving a higher PassRatio [1]. Consequently, for unexecutable codes, the model tends to assign scores approaching zero. In contrast, for codes demonstrating superior functional correctness, the model is likely to assign higher scores. CodeScore has the following advantages: 1) CodeScore has excellent evaluation performance, which achieves the state-of-the-art performance correlation with functional correctness on multiple code evaluation datasets. 2) CodeScore provides three application scenarios (Ref-only, NL-only, and Ref&NL) for code evaluation with unified input, while traditional CEMs only consider Ref-only.

Our major contributions can be summarized as follows: (1) We propose an efficient and effective LLM-based CEM, CodeScore, that accommodates the functional correctness of generated codes from execution viewpoint. (2) We present UniCE, a unified code evaluation learning framework based on LLMs with unified input, which assists models in learning code execution and predicting an estimate of execution PassRatio. (3) We construct three code evaluation datasets based on public benchmark

---

[1]Note that, although PassRatio varies across different test cases, it tends to yield a higher PassRatio for high-quality code, since we generate a large number of test cases. This phenomenon is somewhat akin to the process of human feedback. Despite the inherent variability in scores assigned by different human evaluators, the overarching trend remains consistent.

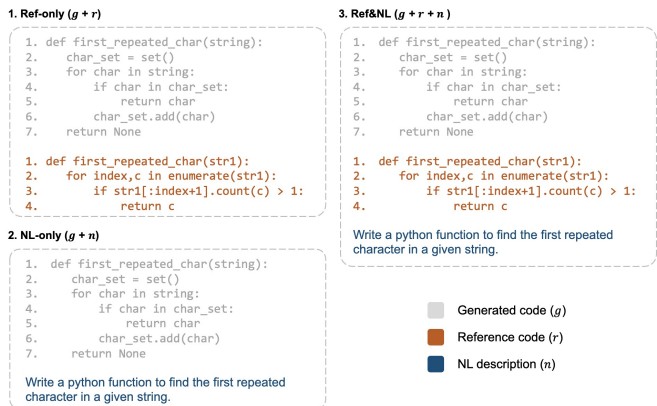

Figure 2: Examples of three input formats for code evaluation.

datasets in code generation, called APPS-Eval, MBPP-Eval, and HE-Eval, respectively. Each task of them contains an NL description, several reference codes, 10+ generated codes, and 100+ test cases. (4) CodeScore substantially outperforms match-based CEMs and achieves the state-of-the-art performance on multiple code evaluation datasets.

## 2 METHODOLOGY

In this section, we first introduce our proposed CEM CodeScore, and then describe a unified code evaluation learning framework (i.e., UniCE), which is used to yield the CodeScore.

### 2.1 CODESCORE

Given an unified input sequence $\mathbf{x}$ that admits the following three types, as shown in Fig. 2:

1. **Ref-only** ($\mathbf{g} + \mathbf{r}$): Generated code concatenated with its reference code,
2. **NL-only** ($\mathbf{g} + \mathbf{n}$): Generated code concatenated with its NL description of requirements.
3. **Ref&NL** ($\mathbf{g} + \mathbf{r} + \mathbf{n}$): Generated code concatenated with both its reference code and NL.

UniCE yields a scalar CodeScore $\in [0, 1]$ and a binary number Exec:

$$\text{CodeScore}, \text{Exec} = \text{UniCE}(\mathbf{x}), \tag{1}$$

where $\text{Exec} = 1$ if $\mathbf{g}$ can be executed successfully with all given test inputs otherwise 0, UniCE is our proposed learning framework, and details of UniCE are presented in Section 2.2.

To correlate predictions of UniCE with code execution, we first collect unified data $U$, then label the data with PassRatio and Executability of $\mathbf{g}$, and finally perform supervised learning with UniCE on the preceding paired data and labels. $U = \{U^i\}_{i=1}^{N}$ contains $N$ triplets, consisting of generated code, reference code, and NL segments. $U^i$ is formed as $(\mathbf{g}^i, \mathbf{r}^i, \mathbf{n}^i)$, where $\forall\, \mathbf{g}^i \neq \epsilon$, and $\forall\, \mathbf{r}^i \cup \mathbf{n}^i \neq \epsilon$. In other words, for each $U^i$, generated code cannot be empty and only one of reference code and NL can be empty.

For a task $p \in P$, let the test case set of $p$ as $C_p = \{(\mathcal{I}_{p,c}, \mathcal{O}_{p,c})\}_{c \in C_p}$, a set of paired test case input $\mathcal{I}_{p,c}$ and test case output $\mathcal{O}_{p,c}$. Although the potential program space can be boundless, test cases permit automatic evaluation of code generation capability. Thus, in contrast to most other text generation tasks, human judgment is unnecessary for code generation. We measure functional correctness with PassRatio, which is defined as

$$\frac{1}{|C_p|} \sum_{c \in C_p} \mathbb{I}\left\{ \text{Eval}\left(\mathbf{g}_p, \mathcal{I}_{p,c}\right) = \mathcal{O}_{p,c} \right\}. \tag{2}$$

where $|\cdot|$ indicates the element number of a set, $\mathbb{I}(\cdot)$ is an indicator function, which outputs 1 if the condition is true and 0 otherwise, and $\text{Eval}\left(\mathbf{g}_p, \mathcal{I}_{p,c}\right)$ represents an evaluation function that obtains outputs of code $\mathbf{g}_p$ by way of executing it with $\mathcal{I}_{p,c}$ as input.

Our framework UniCE can learn existing CEMs, including PassRatio and Passability [2]. In this paper, we choose PassRatio since we want to study execution similarity and continuous PassRatio can better reflect the execution similarity of different codes than binary Passability. In the case of generated code with PassRatio equal to 0, we also use binary Executability to distinguish whether the generated code can be executed successfully with all given test cases, and thus measure its quality.

$$\text{Executability} = \begin{cases} 1, \ if \ code \ is \ executable, \\ 0, \ otherwise. \end{cases} \tag{3}$$

For each $U^i$, we use the preceding metrics to derive its label $L^i$ as $(\text{PassRatio}^i, \text{Executability}^i)$. Dataset $D$ is formed as a set of paired $U^i$ and $L^i$, i.e., $\{(U^i, L^i)\}_{i=1}^N$. We encourage UniCE to learn execution PassRatio by minimizing loss function $\mathcal{L} = \mathcal{L}_C + \mathcal{L}_E$:

$$\mathcal{L} = \mathcal{L}_C + \mathcal{L}_E, \tag{4}$$

$$\mathcal{L}_C = (\text{CodeScore} - \text{PassRatio})^2, \tag{5}$$

$$\mathcal{L}_E = -\log \mathbf{p}(\text{Exec} \mid \text{Executability}), \tag{6}$$

where

$$\mathbf{p}(\text{Exec} \mid \text{Executability}) = \begin{cases} \mathbf{p}(\text{Exec}), \ if \ \text{Executability} = 1, \\ 1 - \mathbf{p}(\text{Exec}), \ otherwise. \end{cases} \tag{7}$$

## 2.2 UNiCE

UniCE relies on LLMs to extract representations of $\mathbf{x}$ and can work with existing pre-trained LLMs, such as CodeBERT Feng et al. (2020) and UniXcoder Guo et al. (2022b). The framework of UniCE is illustrated in Fig. 3.

### 2.2.1 POOLING LAYER

The work Tenney et al. (2019); Zhang et al. (2020); Rei et al. (2020) show that exploiting information from different layers of LLM generally results in superior performance than only the last layer. Therefore, following Peters et al. (2018), we pool information from different layers by using a layer-wise attention mechanism, and the final embedding of a token $t$ can be computed as:

$$e_t = \gamma \sum_{k=1}^{l} e_t^k h^k, \tag{8}$$

where $l$ indicates the number of layers, and $\gamma$ and $h^k$ are trainable weights.

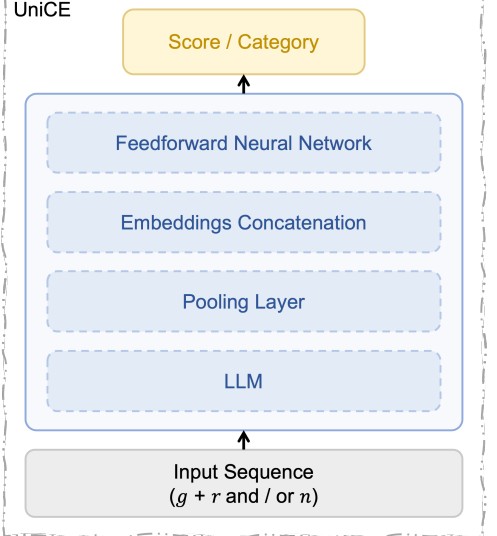

Figure 3: Model architecture of UniCE.

### 2.2.2 UNIFIED EMBEDDING

There are two standard methods to extract total embedding, i.e., averaging all token embeddings and using the first token embedding. Ranasinghe et al. (2020); Wan et al. (2022) proves the superiority of using the first token embedding compared to averaging all token embeddings. Thus, we employ the final embedding of first token $e_{first}$ as the representation of unified input $x$.

### 2.2.3 UNIFIED TRAINING

$e_{first}$ is fed to a feedforward neural network to output a score and/or a category. To unify three evaluation input formats into UniCE, we apply multi-task learning for training. Specifically, for each step, we assign three sub-steps for three input formats, yielding $\mathcal{L}^{Ref}$, $\mathcal{L}^{NL}$, and $\mathcal{L}^{Ref+NL}$,

---

[2] $\frac{1}{|C_p|} \prod_{c \in C_p} \mathbb{I} \{\text{Eval}(\mathbf{g}_p, \mathcal{I}_{p,c}) = \mathcal{O}_{p,c}\}$.

respectively. A Ref&NL data can be regarded as three input format data to yield three losses, while Ref-only and NL-only data can only compute the corresponding $\mathcal{L}^{Ref}$ and $\mathcal{L}^{NL}$. The final learning objective of UniCE is to minimize $\mathcal{L}^{Uni}$:

$$\mathcal{L}^{Uni} = \mathcal{L}^{Ref} + \mathcal{L}^{NL} + \mathcal{L}^{Ref+NL}, \tag{9}$$

where $\mathcal{L}^{Ref}$, $\mathcal{L}^{NL}$, and $\mathcal{L}^{Ref+NL}$ are compute via Eq. 4 using corresponding format data as input.

## 3 EXPERIMENT SETUP

In this section, we introduce datasets, baselines, correlation evaluation, and implementation details. Details of the experiment setup (including datasets and baselines) can be found in Appendix A.

Table 1: Statistics of datasets.

| Dataset | Examples Num | | | Avg Num / Task | | | | Avg Length | | |
|---|---|---|---|---|---|---|---|---|---|---|
| | Train | Dev | Test | NL | RefCode | GenCode | Extended (Original) TestCase | NL | RefCode | GenCode |
| APPS-Eval | 267,162 | 33,395 | 33,395 | 1 | 13 | 42 | 181 (13) | 263.8 | 86.3 | 76.8 |
| MBPP-Eval | 15,679 | 3,000 | 3,000 | 1 | 1 | 24 | 102 (3) | 15.5 | 32.5 | 26.7 |
| HE-Eval | - | - | 1641 | 1 | 1 | 10 | 108 (8) | 61.9 | 24.4 | 47.4 |

### 3.1 DATASETS

We construct three new datasets (named APPS-Eval, MBPP-Eval, and HE-Eval) for code evaluation based on three public benchmark datasets in code generation, i.e., MBPP Austin et al. (2021), APPS Hendrycks et al. (2021), and HumanEval Chen et al. (2021). Statistics of datasets are shown in Table 1. We ensured the correctness of test cases (See Appendix A.1), and then manually filtered some illegal inputs. To avoid data leakage issues in the code evaluation dataset, we ensure that there is no overlap of NL, reference code, and generated code among training, validation, and test sets.

### 3.2 BASELINES

We select typical match-based CEMs and LLM-based EMs as baselines. **Match-based CEMs** include BLEU Papineni et al. (2002), Exact Matching Accuracy (Accuracy), CodeBLEU Ren et al. (2020), and CrystalBLEU Eghbali and Pradel (2022). **LLM-based EMs** contain two well-known and widely used text EMs (BERTScore Zhang et al. (2020) and COMET Rei et al. (2020)) and a recently public CEM (CodeBERTScore Zhou et al. (2023)). The input format of these baselines is Ref-only. Each of the preceding baselines except COMET is in the range of 0 to 1.

### 3.3 CORRELATION EVALUATION

We use three major correlation coefficients in statistics (i.e., Kendall-Tau($\tau$), Spearman R ($r_s$), and Pearson R ($r_p$) to evaluate the correlation between each EM and functional correctness. Furthermore, we use Mean Absolute Error (MAE) to assess the absolute error between them.

**Kendall-Tau** ($\tau$) Kendall (1938) is a statistic used to measure the ordinal association between two measured data:

$$\tau = \frac{Concordant - Discordant}{Concordant + Discordant}, \tag{10}$$

where $Concordant$ indicates the number of occurrences that two evaluation data $M^1$ and $M^2$ exist either both $M_i^1 > M_j^1$ and $M_i^2 > M_j^2$ or both $M_i^1 < M_j^1$ and $M_i^2 < M_j^2$, and $Discordant$ indicates the number of occurrences opposite to $Concordant$.

**Spearman R** ($r_s$) Mood (1950) is a nonparametric measure of rank correlation (statistical dependence between the rankings of two data):

$$r_s = \frac{\mathrm{cov}(\mathrm{R}(M^1), \mathrm{R}(M^2))}{\sigma_{\mathrm{R}(M^1)}\sigma_{\mathrm{R}(M^2)}}, \tag{11}$$

Table 2: Correlation comparison of functional correctness on APPS-Eval.

| Method | Value | $\tau\uparrow$ | $r_s\uparrow$ | $r_p\uparrow$ | MAE $\downarrow$ | Execution Time $\downarrow$ |
|---|---|---|---|---|---|---|
| **Match-based CEM** | | | | | | |
| BLEU | 0.0094 | 0.1055 | 0.1156 | 0.0959 | 0.1164 | $1.0\times$ (26.0s) |
| Accuracy | 0.0001 | 0.0079 | 0.0095 | 0.0196 | - | $0.1\times$ |
| CodeBLEU | 0.2337 | 0.1035 | 0.1533 | 0.1085 | 0.2005 | $7.8\times$ |
| CrystalBLEU | 0.0242 | 0.0906 | 0.1347 | 0.0887 | 0.1709 | $0.3\times$ |
| **LLM-based EM** | | | | | | |
| BERTScore | 0.8629 | 0.0916 | 0.1375 | 0.0718 | 0.6874 | $56.7\times$ |
| COMET | 0.0165 | 0.0904 | 0.1126 | 0.1187 | 0.1751 | $84.0\times$ |
| CodeBERTScore | 0.7583 | 0.1219 | 0.1801 | 0.1323 | 0.5885 | $27.8\times$ |
| **CodeScore** | | | | | | |
| **Ref-only (g + r)** | | | | | | |
| UniCE with $\mathcal{L}^{Ref}$ | 0.1996 | 0.4760 | 0.6473 | 0.6620 | 0.1202 | $33.7\times$ |
| UniCE with $\mathcal{L}^{Uni}$ | 0.1977 | 0.5033 | 0.6693 | 0.6929 | 0.1128 | |
| **NL-only (g + n)** | | | | | | |
| UniCE with $\mathcal{L}^{NL}$ | 0.2035 | 0.4679 | 0.6359 | 0.6855 | 0.1189 | $37.9\times$ |
| UniCE with $\mathcal{L}^{Uni}$ | 0.2016 | 0.4901 | 0.6486 | 0.6905 | 0.1120 | |
| **Ref&NL (g + r + n)** | | | | | | |
| UniCE with $\mathcal{L}^{Ref+NL}$ | 0.1837 | 0.3865 | 0.5419 | 0.6152 | 0.1274 | $44.2\times$ |
| UniCE with $\mathcal{L}^{Uni}$ | 0.1820 | **0.5275** ($\uparrow$ 40.56%) | **0.7040** ($\uparrow$ 55.07%) | **0.7210** ($\uparrow$ 58.87%) | **0.1044** | |

where $R(M^1)$ and $R(M^2)$ represent the rankings of $M^1$ and $M^2$, $\text{cov}(\cdot,\cdot)$ means the covariance function, and $\sigma_M$ means the standard deviation of $M$.

**Pearson R ($r_p$)** Bravais (1844) is a measure of linear correlation between two data:

$$r_s = \frac{\text{cov}(M^1, M^2)}{\sigma_{M^1}\sigma_{M^2}}. \tag{12}$$

**Mean Absolute Error (MAE)** is a measure of errors between paired data:

$$\text{MAE} = \frac{\sum_{i=1}^{N}\left|M_i^1 - M_i^2\right|}{N}, \tag{13}$$

where $|\cdot|$ means the absolute-value function.

### 3.4 IMPLEMENTATION DETAILS

In this paper, UniXcoder Guo et al. (2022b) is employed as the base LLM of UniCE, which has the similar parameter size of LLMs in BERTScore Zhang et al. (2020) and COMET Rei et al. (2020), and larger LLMs can usually lead to better results. We train UniCE with Adam Kingma and Ba (2015) optimizer on a single GPU of Tesla A100-PCIe-40G. Empirically, the learning rate is set to 0.001. The feedforward neural network of UniCE consists of 3 linear transitions with the hyperbolic tangent (Tanh) activation functions, where the corresponding output dimensions are 3,072, 1,024, and 2, respectively. The input token length is limited to 1024. To mitigate the instability of model training, we exhibit the average performance of UniCE running five times.

## 4 EXPERIMENTAL RESULTS

In this section, we conduct extensive experiments to verify the effectiveness and generalization of CodeScore. More comprehensive evaluations and discussions can be found in Appendix B - F.

### 4.1 EFFECT OF CODESCORE

As illustrated in Table 2, CodeScore exhibits a significantly stronger correlation with functional correctness than existing match-based CEMs and LLM-based EMs, which display weak or extremely weak correlations with Ground Truth on APPS-Eval. Compared with the top-performing EM among other EMs, CodeScore achieved absolute improvements of 40.56%, 55.07%, and 58.87% on $\tau$, $r_s$, and $r_p$, respectively. With an $r_s$ value greater than 0.6, it is evident that there is a strong correlation

Table 3: Correlation comparison of functional correctness on MBPP-Eval and HE-Eval.

| Method | MBPP-Eval | | | HE-Eval | | |
|---|---|---|---|---|---|---|
| | Value | $r_s \uparrow$ | Execution Time $\downarrow$ | Value | $r_s \uparrow$ | Execution Time $\downarrow$ |
| **Match-based CEM** | | | | | | |
| BLEU | 0.1186 | 0.1784 | $1.0 \times (0.87s)$ | 0.2249 | 0.0678 | $1.0 \times (0.78s)$ |
| Accuracy | 0.0004 | 0.0299 | $0.1 \times$ | 0.0006 | 0.0367 | $0.1 \times$ |
| CodeBLEU | 0.1827 | 0.2902 | $5.0 \times$ | 0.3826 | 0.4084 | $6.4\times$ |
| CrystalBLEU | 0.0295 | 0.1645 | $0.3 \times$ | 0.0158 | 0.2013 | $0.4 \times$ |
| **LLM-based EM** | | | | | | |
| BERTScore | 0.8842 | 0.1522 | $62.0 \times$ | 0.8862 | 0.0069 | $57.7\times$ |
| COMET | -0.5001 | 0.2681 | $69.0 \times$ | 0.0642 | 0.0716 | $58.6\times$ |
| CodeBERTScore | 0.7863 | 0.2490 | $44.9 \times$ | 0.7917 | 0.2604 | $47.5 \times$ |
| **CodeScore** | | | | | | |
| Ref-only (g + r) | | | | | | |
| UniCE with $\mathcal{L}^{Ref}$ | 0.2975 | 0.5864 | $17.2 \times$ | 0.3115 | 0.5250 | $30.8\times$ |
| UniCE with $\mathcal{L}^{Uni}$ | 0.3253 | 0.5999 | | 0.4055 | 0.6009 | |
| NL-only (g + n) | | | | | | |
| UniCE with $\mathcal{L}^{NL}$ | 0.3364 | 0.4492 | $12.6 \times$ | 0.4748 | 0.5217 | $31.0\times$ |
| UniCE with $\mathcal{L}^{Uni}$ | 0.3327 | 0.5719 | | 0.5357 | 0.5755 | |
| Ref&NL (g + r + n) | | | | | | |
| UniCE with $\mathcal{L}^{Ref+NL}$ | 0.2905 | 0.5926 | $20.7 \times$ | 0.3866 | 0.5153 | $33.3\times$ |
| UniCE with $\mathcal{L}^{Uni}$ | 0.3247 | **0.6054** (↑ 31.52%) | | 0.4505 | **0.6048** (↑ 19.64%) | |

between CodeScore and Ground Truth. Furthermore, CodeScore has the lowest MAE compared to other EMs. The execution time of CodeScore is similar to other LLM-based EMs and slightly longer than existing Match-based CEMs. However, compared to the $20.7k\times$ execution time of execution-based CEMs (reported in Table 8 in Appendix), CodeScore reduces execution time by three orders of magnitude.

We also sought to determine the generality of CodeScore. In Table 3, we utilize CodeScore, trained on APPS-Eval, to evaluate the code in MBPP-Eval and HE-Eval with fine-tuning and zero-shot settings, respectively. It is important to note that these three datasets are quite different, as evidenced by their respective statistics shown in Table 1. Table 3 reveals the effectiveness of CodeScore on MBPP-Eval and HE-Eval. Remarkably, CodeScore continues to achieve the best correlation compared to other EMs in these two settings.

Another intriguing finding is that the quality of CodeBLEU inversely correlates with code length. In other words, the longer code, the poorer correlation between CodeBLEU and Ground Truth. This is likely due to the fact that longer codes tend to incorporate more variations in their syntactic structure. Therefore, for longer codes, the evaluation effect of CodeBLEU gradually degrades to BLEU.

## 4.2 EFFECT OF $\mathcal{L}^{Uni}$

As observed from Tables 2 and 3, our proposed $\mathcal{L}^{Uni}$ demonstrates enhancements across all input formats when compared to their respective losses on APPS-Eval, MBPP-Eval, and HE-Eval datasets. With changes in the input format, both the correlation coefficients and MAE between CodeScore and Ground Truth also vary. Generally, the Ref&NL input format yields superior results, which shows that accommodating NL has a positive effect on evaluating the generated code, while the traditional Ref-only input format omits the valuable information in NL. Additionally, according to the Avg Length data presented in Table 1, we discovered that the execution time of CodeScore exhibits a linear, positive relationship with the input length. Regardless of the input formats, our proposed CodeScore provides a commendable evaluation of generated code. This is attributable to the fact that $\mathcal{L}^{Uni}$ aids in training a code evaluation model with a unified input.

## 4.3 HUMAN EVALUATION

In this section, we conduct a human evaluation to gauge the validity of our CodeScore. Considering the costliness of human evaluation, we select only five representative EMs for this task, namely, CodeScore, CodeBLEU, BERTScore, CodeBERTScore, and Ground Truth (PassRatio). All of these EMs are continuous and range from 0 to 1. In accordance with previous work Hao et al. (2022)

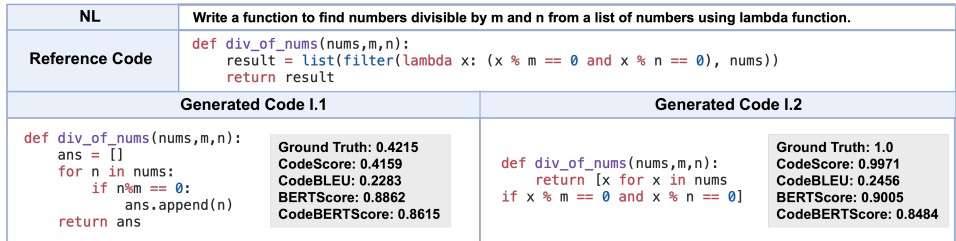

(a) Case I

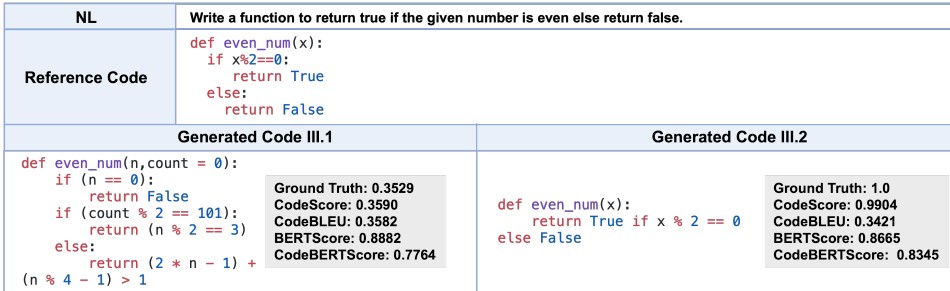

(b) Case II

Figure 4: Case Study on MBPP-Eval.

and our experimental setup, we manually assess the validity of each EM in gauging the functional correctness of the generated code. The score for this evaluation is an integer ranging from 0 to 5, where 0 denotes poor and 5 signifies excellent performance. The details of the human evaluation are outlined in Appendix D.

We present the results of the human evaluation in Table 4. Remarkably, our proposed CodeScore significantly outperforms all other EMs. Relative to these, CodeScore shows an improvement of at least 54.6% in the human evaluation. All p-values are substantially less than 0.005, underscoring that these improvements are statistically significant.

Table 4: Human evaluation for functional correctness.

| EM | Reasonableness |
|---|---|
| BERTScore | 1.3 ± 0.4 |
| CodeBLEU | 2.1 ± 0.5 |
| CodeBERTScore | 2.2 ± 0.7 |
| CodeScore | **3.4 (↑ 54.6%) ± 0.3** |
| Ground Truth | 4.6 ± 0.2 |

## 4.4 CASE STUDY

Fig. 4 displays a selection of generated codes and their corresponding EM scores (as per Section 4.3) on MBPP-Eval. It becomes evident that CodeBLEU, BERTScore, and CodeBERTScore each exhibit unique issues. From these examples, we glean the following insights: 1) CodeBLEU tends to assign relatively low scores to generated code, even when the code is functionally correct. Furthermore, it appears to favor generated codes that maintain structural consistency with the reference code. For instance, even though Generated Code II.2 is functionally correct, it receives a lower CodeBLEU score than II.1, which is fundamentally incorrect. 2) Both BERTScore and CodeBERTScore have a propensity to award relatively high scores to generated code, even when the code is essentially flawed. Additionally, they often assign lower scores to better generated codes. For example, Generated Code II.2 has a lower BERTScore than II.1, and Generated Code I.2 has a lower CodeBERTScore than I.1. In contrast, CodeScore performs admirably in both scenarios. In summary, our proposed CodeScore aligns more closely with Ground Truth compared to other EMs. This suggests that CodeScore is more effective in estimating the functional correctness of generated code.

## 5 DISCUSSION

While we have demonstrated that CodeScore is an effective LLM-based metric for code evaluation, we acknowledge that it still has certain limitations. First, our current version of CodeScore only supports the most popular PL, i.e., Python. Nevertheless, our work establishes the viability of

code evaluation based on UniCE, and this approach can feasibly be extended to other PLs. We aim to broaden CodeScore to encompass multiple PLs in our future work. Second, learning code execution for code evaluation requires collecting a certain amount of data, including sufficient test cases, generated codes, reference codes, and NL descriptions. However, collecting this data is far less expensive than performing human evaluation. Third, employing CodeScore for code evaluation entails additional computation and time. However, we maintain that this is still within an acceptable range, considering the benefits it provides in terms of the accuracy and reliability of code evaluation.

## 6 RELATED WORK

**Match-based CEMs.** Besides these commonly used BLEU Papineni et al. (2002), Accuracy, and CodeBLEU Ren et al. (2020), some niche CEMs Popovic (2015) are also applied to code evaluation, e.g., METEOR Banerjee and Lavie (2005), ROUGE Lin (2004), and CrystalBLEU Eghbali and Pradel (2022). However, these aforementioned match-based CEMs merely measure the surface-level differences in code and do not take into account the functional correctness of the generated code.

**Execution-based CEMs.** They attempt to handle these issues by running tests for generated code to verify its functional correctness Kulal et al. (2019); Hendrycks et al. (2021); Hao et al. (2022). However, they come with several caveats: 1) It assumes that test cases have been given and all dependencies have been resolved. For each code generation task, supplying adequate test cases is a burden in practice, and the dependencies required vary from task to task. 2) Enormous computational overhead needs to be afforded. All generated code requires execution separately for each corresponding test case, which leads to enormous CPU and I/O overhead. 3) Execution with isolation mechanisms. The generated code could have some security risks, such as deleting files on the disk or implanting computer viruses, especially if the training data of code generation models is attacked. In a word, they are usually costly, slow, and insecure, which are often unavailable or ineffective in real-world scenarios.

**LLM-based EMs.** Effective evaluation of generated results is hard for both text and code generation. They likewise face the same issue of poor evaluation metrics (EMs). A recent popular trend in evaluating text generation is the design of automatic EMs based on LLMs. A part of LLM-based EMs Rei et al. (2021); Wan et al. (2022); Rei et al. (2022) follows COMET Rei et al. (2020) to learn high-quality human judgments of training data, which is a problem for code evaluation to obtain. Another part relies on LLM extracting token embeddings to calculate scores like BERTScore Zhang et al. (2020), such as Zhao et al. (2019); Sellam et al. (2020); Yuan et al. (2021); Reimers and Gurevych (2019). They also perform poorly in code evaluation. CodeBERTScore Zhou et al. (2023) tries to use the same way as BERTScore with LLM pre-trained on code. However, simply relying on LLMs to extract the hidden layer to calculate the correlation cannot fundamentally solve the problem that LLMs are confused with how to evaluate code. Therefore, CodeBERTScore does not perform very well in our experiments.

## 7 CONCLUSION AND FUTURE WORK

In this paper, we have proposed a code evaluation learning framework based on LLMs with a unified input, which we refer to as UniCE. UniCe is designed to learn the code execution of generated code. In response to the imprecise evaluations provided by existing match-based CEMs and LLM-based EMs, we introduced CodeScore based on UniCE, which is an effective CEM to measure the functional correctness of generated code. Furthermore, our CodeScore can be applied to three application scenarios (Ref-only, NL-only, and Ref&NL) for code evaluation with a unified input. This is in contrast to traditional CEMs, which typically only consider the Ref-only scenario. To validate CodeScore, we constructed three code evaluation datasets (i.e., APPS-Eval, MBPP-Eval, and HE-Eval), which correspond to three popular benchmark datasets in code generation (i.e., MBPP, APPS, and HumanEval). Experimental results affirm the efficacy of CodeScore, which achieves state-of-the-art performance on multiple code evaluation datasets.

We hope this work sheds light on future work in the direction of LLM-based code evaluation. Our code evaluation dataset can serve as a benchmark for evaluating the functional correctness of generated code. Furthermore, our work can be applied to facilitate the training of code generation models by providing positive feedback.

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
