# APPENDIX

## A   DETAILS OF EXPERIMENTAL SETUP

### A.1   DETAILS OF DATASETS

Table 5: Statistics of datasets (part 2).

| Dataset | AvgPassRatio | | | Pass@1 | | |
|---|---|---|---|---|---|---|
| | Train | Dev | Test | Train | Dev | Test |
| MBPP-Eval | 0.2832 | 0.2571 | 0.2890 | 0.0674 | 0.0494 | 0.0760 |
| APPS-Eval | 0.3196 | 0.1814 | 0.1790 | 0.0315 | 0.0007 | 0.0011 |
| HE-Eval | - | - | 0.3022 | - | - | 0.1499 |

To construct each code evaluation dataset, we first follow primitive NL and reference code in each corresponding base dataset. Then, for each paired NL and reference code in a code evaluation dataset, we generate an average of 20+ codes (generated from various LLMs, including CodeGen 350M&16B Nijkamp et al. (2022), InCoder 1B&6B Fried et al. (2022), and CodeX 13B&175B) Chen et al. (2021) according to NL and additionally build an average of 100+ correct test cases according to reference code. To obtain these test cases, the following steps were implemented:

1) Infer the type of input from pre-existing test cases.

2) Enumerate a collection of inputs constrained by the type of input and task.

3) Feed the input into the original correct code and gets the output by execution (We assume that all external dependencies including third-party libraries have been installed correctly).

Finally, we label each matched NL, reference code, and generated code by executing generated code with all corresponding test cases to compute PassRatio via Eq. 2. Statistics of datasets are shown in Table 5 and Fig. 5.

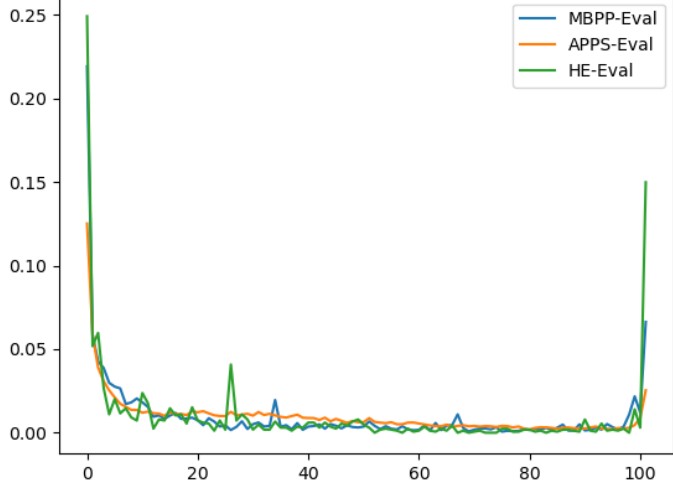

Figure 5: PassRatio distribution of APPS-Eval, MBPP-Eval, and HE-Eval.

## A.2 DETAILS OF BASELINES

We select typical match-based CEMs, LLM-based EMs, and execution-based CEMs as baselines. We present each type of EMs as below.

***Match-based CEMs*** include BLEU Papineni et al. (2002), Exact Matching Accuracy (Accuracy), CodeBLEU Ren et al. (2020), and CrystalBLEU Eghbali and Pradel (2022).

**BLEU** Papineni et al. (2002) is calculated based on n-gram, and the fluency and correctness of generated code are expressed by calculating the proportion of n consecutive tokens in the correct code, where n is usually set to 4 (i.e., BLEU-4). Considering that shorter codes usually have higher BLEU values, a penalty item is introduced to BLEU as:

$$\text{BLEU} = BP \cdot \exp \left( \sum_{m=1}^{n} \omega_m \log p_m \right),$$

$$BP = \begin{cases} 1, & l_g \geq l_r \\ e^{\left\{ 1 - \frac{r}{l_g} \right\}}, & l_g < l_r \end{cases},$$

where $BP$ represents the penalty item, $l_g$ represents the length of generated code, $l_r$ represents the length of reference code, and $\omega_m$ and $p_m$ represents the weighted coefficient and precision of $m$-gram, respectively.

**Accuracy** indicates the percentage of exact matches between generated code and reference code.

**CodeBLEU** Ren et al. (2020) additionally takes into account the structure of code, which absorbs the advantages of BLEU in n-gram matching, and further injects code syntax through abstract syntax tree and code semantics through data flow.

$$\text{CodeBLEU} = \alpha \cdot \text{BLEU} + \beta \cdot \text{BLEU}_{weight} + \delta \cdot \text{Match}_{ast} + \zeta \cdot \text{Match}_{df},$$

where $\alpha, \beta, \delta$ and $\zeta$ are weights (usually set to 0.25, as well as in this paper), $\text{BLEU}_{weight}$ is a weighted BLEU with different weights for various tokens, $\text{Match}_{ast}$ is syntactic AST matching, which explores the syntactic information of the code, and $\text{Match}_{df}$ is semantic dataflow matching, which considers the semantic similarity between generated code and reference code.

**CrystalBLEU** Eghbali and Pradel (2022) is a metric that calculates BLEU by reducing the noise caused by trivially shared n-grams, such as '(' and ','.

***LLM-based EMs*** contain BERTScore Zhang et al. (2020) and COMET Rei et al. (2020), which are well-known and widely used.

**BERTScore** Zhang et al. (2020) is an automatic evaluation metric for text generation, which computes a similarity score for each token in the generated sentence with each token in the reference sentence with contextual embeddings of BERT Devlin et al. (2019).

$$R_{\text{BERT}} = \frac{1}{|\mathbf{x}|} \sum_{\mathbf{x}_i \in \mathbf{x}} \max_{\hat{\mathbf{x}}_j \in \hat{\mathbf{x}}} \mathbf{x}_i^\top \hat{\mathbf{x}}_j, \quad P_{\text{BERT}} = \frac{1}{|\hat{\mathbf{x}}|} \sum_{\hat{\mathbf{x}}_j \in \hat{\mathbf{x}}} \max_{\mathbf{x}_i \in \mathbf{x}} \mathbf{x}_i^\top \hat{\mathbf{x}}_j,$$

$$F_{\text{BERT}} = 2 \frac{P_{\text{BERT}} \cdot R_{\text{BERT}}}{P_{\text{BERT}} + R_{\text{BERT}}}.$$

Following the setting in Zhang et al. (2020), we compute BERTScore with inverse document frequency computed from test sets.

**COMET** Rei et al. (2020) provides a text EM by learning human judgments of training data, which leverages cross-lingual pre-trained language modeling to predict the quality of generated text more accurately.

***Execution-based CEMs*** as the gold standards consist of **AvgPassRatio** Hendrycks et al. (2021) and **Pass@1** Kulal et al. (2019), which are computed via Eq. 14 and Eq. 15, respectively.

Each of the preceding baselines except COMET is in the range of 0 to 1.

## B  EFFECT OF BINARY CODESCORE

In Table 6, we convert all EMs to binary EMs using thresholds picked in a similar way to binary CodeScore, and we compare their correlation with Pass@1 on three code evaluation datasets. As we can see, our Binary CodeScore still exhibits the best correlation with Ground Truth, compared to binary match-based CEMs and LLM-based EMs. Specifically, binary CodeScore is moderately correlated with Ground Truth, while binary match-based CEMs and LLM-based EMs have weak or extremely weak correlations with Ground Truth@. Moreover, the execution time of binary EMs does not increase significantly and is almost equal to EMs. Thus, we do not report it additionally, and it can refer to the execution time of EM in Table 2 and Table 3.

Table 6: Correlation comparison of functional correctness with binary EMs on code evaluation datasets, where three correlation coefficients are equal (i.e., $\tau = r_s = r_p$) for two Bernoulli distributed data.

| Method (Binary) | MBPP-Eval | APPS-Eval | HE-Eval |
|---|---|---|---|
| | $r_s \uparrow$ | $r_s \uparrow$ | $r_s \uparrow$ |
| **Match-based CEM** | | | |
| Binary BLEU | 0.0928 | 0.0907 | 0.0887 |
| Accuracy | 0.0636 | 0.0369 | 0.0586 |
| Binary CodeBLEU | 0.2059 | 0.1666 | 0.2409 |
| **LLM-based EM** | | | |
| Binary BERTScore | 0.0972 | 0.0851 | 0.0375 |
| Binary COMET | 0.2358 | 0.1897 | 0.1181 |
| **CodeScore** | | | |
| Binary CodeScore | **0.4164** | **0.4176** | **0.4159** |

Table 7: Comparison of CodeScore (trained on APPS-Eval) and the SOTA reranking methods for code generation, where PTC means filtering with a public test case after reranking. We use italics for results reported from reference papers.

| Model | MBPP | | | HumanEval | | |
|---|---|---|---|---|---|---|
| | Pass@1 ↑ | Pass@1 (ET) ↑ | Reranking Time ↓ | Pass@1 ↑ | Pass@1 (ET) ↑ | Reranking Time ↓ |
| CodeX 175B Chen et al. (2021) | *0.581 (0.576)* | 0.388 | - | *0.470 (0.445)* | 0.317 | - |
| + MBR-EXEC Shi et al. (2022) | *0.639* | - | 56.4 × | *0.505* | - | 53.5 × |
| + CodeT Chen et al. (2022) | ***0.677*** | 0.451 | 123.1 × | ***0.658*** | **0.517** | 116.6 × |
| + CodeScore | 0.627 | **0.472** | 1.0 × (34.2s) | 0.516 | 0.409 | 1.0 × (24.0s) |
| + CodeScore + PTC | 0.858 | 0.677 | 3.0 × | 0.636 | 0.520 | 2.9 × |

## C  CODESCORE FOR RERANKING

We compare CodeScore (trained on APPS-Eval) and the state-of-the-art (SOTA) reranking methods of code generation, including MBR-EXEC Shi et al. (2022) and CodeT Chen et al. (2022), on two code generation benchmark datasets using Pass@1. We first use Exec via Eq. 1 to filter codes that can be executed successfully and then employ CodeScore to rank the remaining codes with NL-only input format. In Table 7, reranking with CodeScore improves in all cases, although it is still lower than CodeT in some cases. However, considering that CodeT is difficult to apply to practical scenarios due to huge time and calculation overhead, our CodeScore takes advantage. Moreover, we find that the public test case (PTC) can further cause a huge improvement, where PTC is randomly selected from self-contained PTC or extended test cases and has no intersection with private original test cases. Compared with Pass@1, Pass@1 (ET) on both datasets has a significant decline, indicating that original test cases cannot adequately measure the functional correctness of generated code. However, reranking methods with CodeScore significantly mitigate degradation compared to base code generation model and '+ CodeT Chen et al. (2022)', which means that CodeScore has a more comprehensive assessment of the code's functional correctness.

# D    DETAILS OF HUMAN EVALUATION

We randomly select 100 generated codes and evaluations scored by the five EMs mentioned above on these samples. Finally, we obtain 500 (100*5) paired generated code with reference code and NL and its EM score for human evaluation. The evaluators are computer science Ph.D. students and are not co-authors. They all have programming experience ranging from 3+ years. The 500 code snippets are divided into 10 groups, with each questionnaire containing one group. We randomly list the generated code with reference code and NL and the corresponding EM score on the questionnaire. Each group is evaluated anonymously by one evaluator, and the final score is the average of all evaluators' scores. Evaluators are allowed to search the Internet for unfamiliar concepts.

Table 8: comparison of CodeScore and other EMs on APPS-Eval.

| Method | Value | $\tau \uparrow$ | $r_s \uparrow$ | $r_p \uparrow$ | MAE $\downarrow$ | Execution Time $\downarrow$ |
|---|---|---|---|---|---|---|
| **Match-based CEM** | | | | | | |
| BLEU | 0.0094 | 0.1055 | 0.1156 | 0.0959 | 0.1164 | $1.0 \times$ (26.0s) |
| Accuracy | 0.0001 | 0.0079 | 0.0095 | 0.0196 | - | $0.1 \times$ |
| CodeBLEU | 0.2337 | 0.1035 | 0.1533 | 0.1085 | 0.2005 | $7.8 \times$ |
| CrystalBLEU | 0.0242 | 0.0906 | 0.1347 | 0.0887 | 0.1709 | $0.3 \times$ |
| **LLM-based EM** | | | | | | |
| BERTScore | 0.8629 | 0.0916 | 0.1375 | 0.0718 | 0.6874 | $56.7 \times$ |
| COMET | 0.0165 | 0.0904 | 0.1126 | 0.1187 | 0.1751 | $84.0 \times$ |
| CodeBertScore | 0.7583 | 0.1219 | 0.1801 | 0.1323 | 0.5885 | $27.8 \times$ |
| **Execution-based CEM** | | | | | | |
| AvgPassRatio | 0.0978 | 0.3360 | 0.4108 | 0.4987 | 0.1327 | $1.5\text{k} \times$ |
| Pass@1 | *0.0064* | 0.0894 | 0.1983 | 0.1598 | - | |
| Pass@1 (ET) | *0.0011* (↓ 82.8%) | 0.0470 | 0.0569 | 0.1174 | - | $20.7\text{k} \times$ |
| UniCE | 0.1820 | **0.5275** | **0.7040** | **0.7210** | **0.1044** | $44.2 \times$ |

# E    COMPARISON WITH EXECUTION-BASED CEMS

In Table 8, we can find that computing execution-based CEMs for code evaluation with a small number of original test cases is insufficient. They have a significant reduction in correlation coefficients compared to using larger extended test cases. The value of Pass@1 (ET) is 82.8% lower than Pass@1, meaning that 82.8% of correct codes under Pass@1 are misjudged on APPS. In cases where test cases are rare or low-quality, such as on APPS-Eval, the correlation between our CodeScore and Ground Truth even far exceeds that of AvgRassRatio. Therefore, the quality of AvgRassRatio's evaluation depends on the quality and quantity of test cases.

In conclusion, AvgPassPatio and Pass@1 are effective but costly, since computing AvgPassPatio and Pass@1 not only requires adequate test samples but also dramatically increases the execution time. Therefore, in the situation of insufficient test cases, our CodeScore is a suitable alternative to them for quality and cost.

# F    ADDITIONAL CASE STUDY

As shown in 6, CodeBLEU tends to assign relatively low scores to generated code, even when the code is functionally correct. Both BERTScore and CodeBERTScore tend to award relatively high scores to generated code, even when the code is essentially flawed. Additionally, BERTScore often assigns lower scores to the more functional correctness generated codes. For example, Generated Code III.2 has a lower BERTScore than III.1. In contrast, CodeScore performs admirably in this scenario. In summary, our proposed CodeScore aligns more closely with Ground Truth compared to other EMs. This suggests that CodeScore is more effective in estimating the functional correctness of generated code.

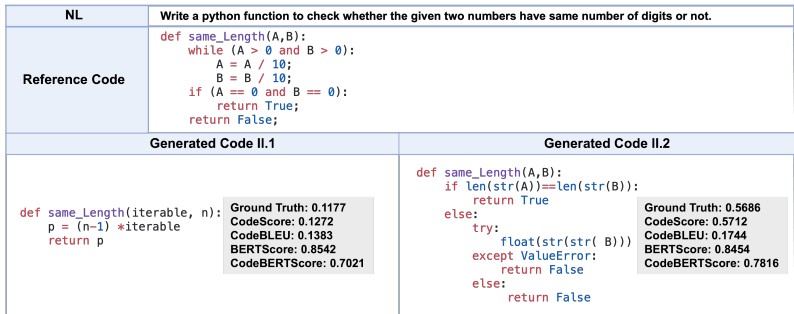

Figure 6: Case III

## G  PRELIMINARY KNOWLEDGE

For a task $p \in P$, let the test case set of $p$ as $C_p = \{(\mathcal{I}_{p,c}, \mathcal{O}_{p,c})\}_{c \in C_p}$, a set of paired test case input $\mathcal{I}_{p,c}$ and test case output $\mathcal{O}_{p,c}$. Although the potential program space can be boundless, test cases permit automatic evaluation of code generation capability. Thus, in contrast to most other text generation tasks, human judgment is unnecessary for code generation. As the gold standards for code evaluation, AvgPassRatio and Pass@1 mirror the performance of generated codes on test cases.

**AvgPassRatio** The average proportion of test cases that generated codes $\mathbf{g}'_p s$ pass:

$$\frac{1}{|P|} \sum_{p \in P} \frac{1}{|C_p|} \sum_{c \in C_p} \mathbb{I}\{\mathrm{Eval}(\mathbf{g}_p, \mathcal{I}_{p,c}) = \mathcal{O}_{p,c}\}, \qquad (14)$$

where $|\cdot|$ indicates the element number of a set, $\mathbb{I}(\cdot)$ is an indicator function, which outputs 1 if the condition is true and 0 otherwise, and $\mathrm{Eval}(\mathbf{g}_p, \mathcal{I}_{p,c})$ represents an evaluation function that obtains outputs of code $\mathbf{g}_p$ by way of executing it with $\mathcal{I}_{p,c}$ as input.

**Pass@1** The percentage of $\mathbf{g}'_p s$ that pass all test cases of the corresponding $p$:

$$\frac{1}{|P|} \sum_{p \in P} \frac{1}{|C_p|} \prod_{c \in C_p} \mathbb{I}\{\mathrm{Eval}(\mathbf{g}_p, \mathcal{I}_{p,c}) = \mathcal{O}_{p,c}\}. \qquad (15)$$

Pass@1 is a more stringent CEM, which is a representative of the Pass@k family, also known as Strict Accurracy. In this paper, we use original and extended test cases to compute Pass@1 and Pass@1 (ET), respectively.

## H  TEST CASE GENERATION VIA CHATGPT

We randomly select 100 code generation tasks from the MBPP dataset and use the NL description and reference code of tasks to generate test cases via ChatGPT OpenAI. Fig. 7 shows an example of ChatGPT generating test cases. ChatGPT generates an average of 1.53 test cases per task.

The results shown in Fig. 8 indicate that **LLMs have the potential to judge the functional correctness of most programs with appropriate guidance**. Only 1.29% Generations consistent with private test cases means that ChatGPT generates test cases by itself instead of copying private test cases.

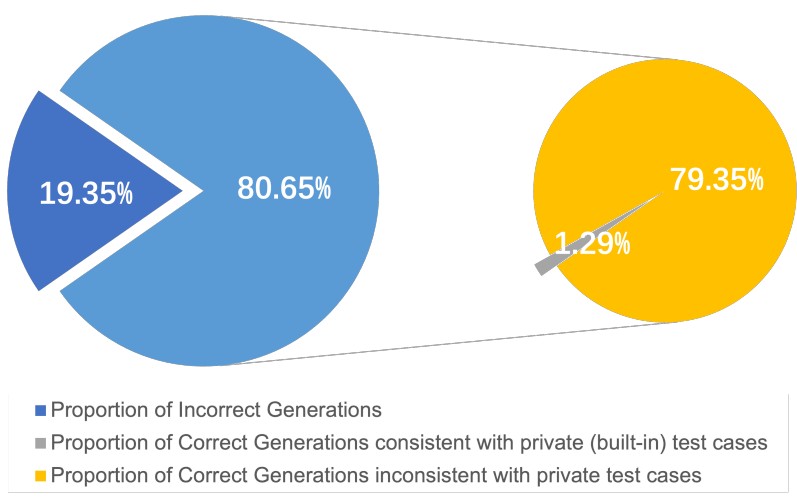

```
# Write a python function to find the first repeated character in a given string.
def first_repeated_char(str1):
  for index,c in enumerate(str1):
    if str1[:index+1].count(c) > 1:
      return c

# test the function with some example inputs
```

```
assert first_repeated_char("abcdefg") == None
assert first_repeated_char("abcdabcd") == "a"
assert first_repeated_char("abcdcabcd") == "c"
```

Figure 7: Example of ChatGPT generating test cases.

19.35%   80.65%

79.35%

1.29%

■ Proportion of Incorrect Generations
■ Proportion of Correct Generations consistent with private (built-in) test cases
■ Proportion of Correct Generations inconsistent with private test cases

Figure 8: Test case generation via ChatGPT OpenAI in zero-shot setting (details can be found in Appendix H).