# OpenReview forum: "CodeScore: Evaluating Code Generation by Learning Code Execution"
_ICLR.cc/2024/Conference — ICLR 2024 Conference Withdrawn Submission_

### Official Review · Reviewer_oV3N · 2023-10-29

**Soundness:** 2 fair
**Presentation:** 2 fair
**Contribution:** 2 fair
**Rating:** 5
**Confidence:** 4

**Summary:**

- This paper presents CodeScore, an LLM-based evaluator for code.
- This work utilizes code embeddings from the UnixCoder model and trains additional layers on top of it to predict two metrics: CodeScore $\in [0, 1]$ a continuous metric, and Executabilty $\in \{0, 1\}$ a binary metric indicating whether the code can be executed or not
- To train the model, the work utilizes existing code generation benchmarks such as APPS, MBPP, and HumanEval and extends them with additional code samples and test cases. Additional code samples are generated from other LLMs, while test cases are generated by inferring the input types from existing correct implementations, creating inputs that conform to the input types, and then passing them to the correct implementation to get reference output, thus creating input and output pairs.
- The model is then trained in a supervised manner with the input as the generated code, the reference code, and the NL description to predict the pass ratio (fraction of test cases the code will pass) and executability (if all test cases will pass or not).
- Testing this method on three benchmarks - APPS, MBPP, and HumanEval, the work shows a higher correlation with functional correctness for CodeScore as compared to other fuzzy metrics such as CodeBLEU and CodeBERTScore.

**Strengths:**

- This paper aims to solve a critical problem in code generation by presenting a metric to evaluate generated code.
- The experiments provide a good overview of the available baselines and the correlation along with the execution time.
- Overall, the paper is well-written and easy to understand.

**Weaknesses:**

- Learning code executability is not novel and has been explored in a similar technique by [1] to improve code generation performance. This work does not mention [1] and does not distinguish between the two techniques.
- This work successfully demonstrates that code execution can be learned (to a certain degree) by deep networks utilizing embeddings from pre-trained code models, but I am not convinced that this method can be used as an evaluation metric for the following reason: a critical problem in code evaluation is the lack of test cases to evaluate code. This paper proposes a simple way of generating test cases by inferring input data type, generating input data of the particular type, and then generating expected output by executing the reference code. This procedure limits the applicability of the metric in terms of the complexity of code that it can evaluate because it is not clear how to generate test data for more complex cases, such as functions that do not return anything (they modify variables in place or alter the state). This is not a desired feature of a metric as it should be able to evaluate a broad range of inputs. For instance, this method cannot be applied to the CoNaLa benchmark (another code generation benchmark) [2].


**Minor comments:**
- The citations should be parenthesized. For example: `Some commonly used match-based CEMs treat code as text, such as BLEU Papineni et al. (2002) and Accuracy`, should be `Some commonly used match-based CEMs treat code as text, such as BLEU (Papineni et al. (2002)) and Accuracy`
- In Section 2.1, the paper mentions "human judgment is unnecessary for code generation". I differ from this statement as functional correctness is only one aspect of code. Other aspects that require human judgment include readability and maintainability of code, among others.
- The last rows of Tables 2 and 3 do not align with the columns making them hard to read.



[1] - Ni, Ansong, et al. "Lever: Learning to verify language-to-code generation with execution." International Conference on Machine Learning. PMLR, 2023.

[2] - https://conala-corpus.github.io/

**Questions:**

Listed above.

---

### Official Review · Reviewer_2x5Y · 2023-10-31

**Soundness:** 2 fair
**Presentation:** 3 good
**Contribution:** 3 good
**Rating:** 3
**Confidence:** 5

**Summary:**

This paper introduces CodeScore, an LLM-based code evaluation metric (CEM) designed to estimate the functional correctness of generated code, transcending mere token and structural similarity.

**Strengths:**

* The newly proposed CEM is adaptable to three input formats (reference-only, NL-only, reference+NL), addressing a pressing need in current CodeLLM research.
* Unlike other metrics that primarily focus on token and structural similarity, this CEM emphasizes functional correctness and the identification of code equivalence.

**Weaknesses:**

A practical enhancement would be to deploy CodeScore in evaluating and ranking the performance of existing CodeLLMs, comparing these rankings against established benchmarks, such as HumanEval. Such experiments have previously been undertaken, as seen in the CodeBLEU paper.

**Questions:**

Questions:
Regarding Table 8, why is there such a low correlation score between Pass@1 and functional correctness?

Recommendations:

Given the metric's proficiency in gauging functional correctness and pinpointing code equivalence, I recommend further assessments using Code Clone Detection tasks, such as BigCloneBench [1] and SemanticCloneBench [2]. This could provide additional depth and validation to the study.

[1] Towards a Big Data Curated Benchmark of Inter-Project Code Clones
[2] Can Neural Clone Detection Generalize to Unseen Functionalities?

---

### Official Review · Reviewer_kAYJ · 2023-11-01

**Soundness:** 2 fair
**Presentation:** 2 fair
**Contribution:** 2 fair
**Rating:** 3
**Confidence:** 4

**Summary:**

This paper targets addressing the limitations of existing code evaluation metrics (CEMs) used in code generation model evaluation. Metrics like BLEU and CodeBLEU primarily measure surface differences between code examples and are designed for a single type of input format (Ref-only), neglecting functional equivalence and versatility. To overcome these limitations, the paper introduces CodeScore based on large language models (LLMs), estimating the functional correctness of generated code across three input types: Ref-only, NL-only, and Ref&NL. To obtain CodeScore, the paper presents UniCE, a unified code generation learning framework that enables LLMs to learn code execution, including PassRatio and Executability, with unified input. Experimental results on multiple code evaluation datasets demonstrate that CodeScore improves correlation with functional correctness compared to static CEM such as BLEU score, achieving state-of-the-art performance and effectively handling all three input formats. It addresses the need for a more comprehensive and versatile code evaluation metric in the context of code generation research.

**Strengths:**

+ It is an interesting and novel direction to evaluate the code functional correctness by estimating the execution results, rather than really executing the programs.
+ The evaluation illustrates CodeScore's effectiveness over static CEM, including the match-based CEM and the LLM-based CEM.

**Weaknesses:**

__The fundamental flaw in UniCE is that it trains the model to estimate the test case pass ratio without showing the test cases themselves.__

The design of UniCE to predict the test case pass ratio without feeding the test cases themselves as the model input is problematic. As explained in Section 2.1, the ground truth of pass ratio score is collected by executing the test cases with specified input and verifying the correctness with the expected output. However, from the description of the rest of this and following sections, it sounds like the test cases will only be used to compute the ground truth score while not being fed into the model as input, since only Ref, NL or their combination will be fed. There are two main flaws in such a design.

First, the test case pass ratio is not a deterministic ground-truth score and will be unstable when the test cases are updated. Concretely, by the definition, such score will vary according to the quality and quantity of the test cases for a specific code sample and creating more difficult test cases and re-running the execution tends to lower the score and vice versa. This is problematic that, even the training code samples are unchanged, if test cases are updated, the model needs to be trained again.

Second, the model does not have enough resources to reason about the real execution when the test cases are not present. Even for an experienced human expert in programming, it is not possible to tell the likelihood of isolated code snippets passing some unknown test cases. UniCE's design actually throws a question mark on whether the model really understands and estimates the execution or they are simply matching a code sequence to some scores with unknown meanings.

Therefore, I would urge the authors to justify the design of UniCE that why the model is supposed to predict the test case pass ratio without even seeing the test cases themselves.

__No evidence is provided in the paper that UniCE learns to estimate the program execution as designed.__

Learning program execution has been widely recognized as a harder task than predicting code tokens, due to the unpredictability of dynamic program behaviors and the complicated, invisible semantics during the execution, such as the complex arithmetic calculation and the loops and recursions. It is not discussed in the paper what perspective and to what extent, UniCE has learned to estimate the program execution, or if the model simply memorizes a mapping function between the input sequence and a specific score rather than reasoning about the execution in depth. I would encourage the authors to probe UniCE a bit to verify whether UniCE really learns to estimate the program execution.

__The additional benefits brought by CodeScore over execution-based CEM are limited__

In the related work (Section 6), the authors claim three drawbacks of execution-based CEMs: (1) the requirement of existing test cases (2) computation overhead for execution (3) the potential risk of triggering security flaws. While I agree CodeScore could alleviate such drawbacks during the inference, CodeScore also has three limitations (1) CodeScore's accuracy compared to execution-based CEM is strictly lower (2) CodeScore loses the explainability of execution-based CEMs (3) CodeScore maintains all the drawbacks of execution-based CEMs during training UniCE. For the first limitation, since UniCE is trained to predict the execution-based CEM as ground truths, the latter is the upper bound of the former approach. For the second limitation, please see the weaknesses below. For the third limitation, as explained in the paper, to train UniCE, all training samples have to be executed with test cases, so all the risks and costs of execution-based CEM remain when training UniCE

While, conceptually, there might be no clear winners between execution-based CEM and CodeScore, and it seems to be a tradeoff of performance and costs when deciding to use which one, it is arguably more important to weigh accuracy and explainability when evaluating the code generation models. Therefore, CodeScore could not bring notable benefits comparing to the popular execution-based CEM, such as Pass@K.

__CodeScore could not maintain the explainability of execution-based CEM.__

Beyond just a score to evaluate the model's capability in code generation, execution-based CEM typically serves as a feedback mechanism to underscore the weaknesses of the model being evaluated. For example, the model developer could log the execution of test cases and easily locate which test cases fail while the other pass. Such information helps the developer pinpoint the issues and weaknesses of models in generating code and consequently, they could further improve that. As a static estimation of execution-based CEM, CodeScore seems to not only sacrifice the accuracy of execution-based CEMs to gain lower evaluation overhead but also the explainability of why a low score is being produced by the model. This further weakens the contribution of CodeScore.

**Questions:**

- Can the authors explain what is the assumption that the UniCE is supposed to accurately estimate the execution pass rate even without seeing the test cases or testing inputs themselves?

---

### Official Review · Reviewer_hATV · 2023-11-03

**Soundness:** 2 fair
**Presentation:** 2 fair
**Contribution:** 2 fair
**Rating:** 3
**Confidence:** 3

**Summary:**

This paper introduces a trainable model, CodeScore, for predicting the functional correctness of generated code functions, using a reference function, a natural language description of the code, or both. To train and evaluate CodeScore, the paper generates additional test cases for functions from the APPS, MBPP, and HumanEval datasets, and generates candidate code solutions from LLMs. CodeScore is trained to predict the ratio of test cases that code solutions pass, and whether the code solutions are executable. The paper evaluates CodeScore on held-out functions (I believe) from the datasets, and shows that it has higher correlation with a measure of functional correctness than past evaluation metrics.

**Strengths:**

S1) A flexible framework that can be both reference-based or reference-free (using only natural language description of the code), and interesting experiments comparing the effectiveness of NL vs Ref vs NL+Ref.

S2) Highlights important issues, that functional correctness and surface-level match to reference code are often unrelated to each other.

S3) The generated test cases for APPS, MBPP, and HumanEval might potentially be useful for other work (but see questions).

**Weaknesses:**

W1) I found the evaluation limited:

W1a) My biggest concern is that (unlike previous training-free frameworks such as CodeBERTScore and match-based CEM), this approach requires executable test cases as training data, which could cause a domain mismatch when evaluating functions from other domains. I do appreciate that the paper did experiments that trained on APPS and evaluated on HumanEval, but it would really benefit the paper to evaluate on another naturalistic NL->code generation benchmark with executable test cases, such as DS-1000 or ODEX.

W1b) The baselines being compared to are a bit weak. BERTScore and COMET were designed for text, not code (and COMET was designed for machine translation!). CodeBERTScore compares representations from a code-pretrained encoder, but is not trained to produce snippet-level representations (it was continued-pretrained with an MLM objective, and uses alignment. I recommend the paper tries one or both of the following baselines:
- Use CodeBERT or another contrastively-trained NL <> code retrieval model comparison in the (g+n) setting zero-shot, and possibly also fine-tuned as UniCE was fine-tuned.
- Use NL as an additional input to CodeBERTScore, as the model was designed for.

W1c) If I understand correctly (see questions below), past work (Zhou et al., CodeBERTScore) uses a different binary-valued definition of functional correctness, while this paper uses a scalar-valued PassRatio functional correctness, determined using generated test-cases. This complicates the comparison to past work (Zhou et al. report much higher correlations for their method on HE than are reported in this paper), and I'm not sure what the justification for it is (see W2).

W2) It's unclear to me how meaningful it is to predict a scalar-valued measure of functional correctness derived via the input-generation methods used in this paper (A.1). It seems that the results of many test cases are correlated with each other (as shown in Fig 5, it looks like a large number of instances have PassRatios of either 0 or 1), and it's unclear whether this is because the test inputs are simple or just because functional correctness tends to be binary. The examples in Fig 4 with values around 0.3 and 0.4 are helpful, but it would help to do more to support this metric.

W3) Parts of the experimental methodology (and to a lesser degree, the model) were pretty unclear to me. See questions below, but my main points of confusion were around:
- The prediction and evaluation of the Executability score Exec.
- The data that the models were trained on.
- The "binary CodeScore" experiments.

**Questions:**

Q1) What data were the UniCE models in Table 1 trained on (APPS-Eval train set, or APPS-Eval+MBPP-Eval train sets, or something else)?

Q2) Is "functional correctness" in the results tables the [0, 1]-valued PassRatio defined in (2), or a binary variable that indicates whether the code passes *all* test cases (which is a commonly-used definition of "functional correctness", as used in the CodeBERTScore paper).

Q3) In Table 8, the correlations for Pass@1 (ET) are extremely low. If I'm understanding right, Pass@1 (ET) has a spearman correlation of 0.0579 with PassRatio -- is that correct? Assuming that pass@1 is a good code evaluation metric (which I believe), to me this means that we shouldn't be using spearman correlation to PassRatio as an evaluation metric for CEM! This is a pretty important question, as to me it calls into question the whole evaluation methodology of the paper.

Q4) Do any results in the paper evaluate the "Exec" number predicted in (1)?

Q5) What is the "binary CodeScore" mentioned in B? I was unclear whether this is Exec, or something else.

Q6) "To mitigate the instability of model training, we exhibit the average performance of UniCE running five times": does this mean the model was trained fine times, with inference results averaged across these five models?

Minor points
- \citep (parenthesize author names) should be used for most citations
- Figure 1 would be stronger if it also showed the code scores predicted by CodeScore  on these examples. It's also unclear to me why the BLEU and CodeBLEU of the reference are <1.
- It was surprising that Ref & NL generally performs worse than NL-only, and it would be interesting to see some discussion of this.